# Insights into the Oral Bacterial Microbiota of Sows

**DOI:** 10.3390/microorganisms9112314

**Published:** 2021-11-09

**Authors:** Jasmine Hattab, Giuseppe Marruchella, Alberto Pallavicini, Fabrizia Gionechetti, Francesco Mosca, Abigail Rose Trachtman, Laura Lanci, Luigino Gabrielli, Pietro Giorgio Tiscar

**Affiliations:** 1Faculty of Veterinary Medicine, University of Teramo, 64100 Teramo, Italy; jhattab@unite.it (J.H.); gmarruchella@unite.it (G.M.); fmosca@unite.it (F.M.); artrachtman@unite.it (A.R.T.); laura.lanci@studenti.unite.it (L.L.); 2Department of Life Sciences, University of Trieste, 34127 Trieste, Italy; pallavic@units.it (A.P.); fgionechetti@units.it (F.G.); 3Veterinary Practitioner, 63073 Ascoli Piceno, Italy; luiginogabrielli@gmail.com

**Keywords:** oral bacterial microbiota, sow, saliva, oral fluid

## Abstract

The investigation of bacterial microbiota represents a developing research field in veterinary medicine intended to look for correlations between animal health and the balance within bacterial populations. The aim of the present work was to define the bacterial microbiota of the oral cavity of healthy sows, which had not been thoroughly described so far. In total, 22 samples of oral fluid were collected and analyzed by 16S-rRNA gene sequencing. CLC Genomics Workbench 20.0 (QIAGEN Digital Insights, Aarhus, Denmark) was then used to examine the results. The predominant orders were Lactobacillales, Clostridiales, and Corynebacteriales. Lactobacillaceae, Corynebacteriaceae, Moraxellaceae, Aerococcaceae, and Staphylococcaceae were the most represented families. As regards the most abundant genera, *Lactobacillus*, *Corynebacterium*, *Acinetobacter*, *Staphylococcus*, *Rothia*, *Aerococcus*, and *Clostridium* can be pointed out as the bacterial core microbiota. Sows were also divided into “gestating” and “lactating” groups, and mild differences were found between pregnant and lactating sows. The data herein described represent an original contribution to the knowledge of the porcine bacterial microbiota. Moreover, the choice of sows as experimental animals was strategic for identifying the adult microbial community. These data provide a basis for further studies on the oral bacterial microbiota of pigs.

## 1. Introduction

The microbiota is the community composed of all the living members of a microbiome, namely bacteria, archaea, fungi, algae, and small protists. The microbiota, together with the microbial structures, metabolites, and mobile genetic and relic DNA embedded in the environmental conditions of the habitat, constitute the microbiome [1,2].

The development of bacterial diseases is often due to alterations of the “dynamic balance” of species sharing the same ecosystem; in fact, many potential pathogens can be present in a host without causing disease. Thus, the knowledge of the interactions among resident—but also between resident and “foreign”—bacteria is crucial to understand the patterns that induce disease [3].

Exploring microbial diversity was a challenge until very few years ago, being limited to culture-dependent methods [4]. Rather new techniques such as 16S rRNA gene amplicon sequencing allowed researchers to identify the components of a microbial community to the genus level. This method is based on the similarities and differences of a single gene and its hypervariable region (V4) to available genome databases [5].

In humans, the oral bacterial microbiota was thoroughly investigated, and, interestingly, most of the studies carried out thereon in association with specific conditions led to greater understanding about oral tumors, diabetes, and oral hygiene [4]. Actually, the salivary bacterial microbiota in humans displays inter-individual differences, although people belonging to the same populations share similar microorganism proportions and composition [6]. The alteration of the known “normal” microbiota characterizing the saliva of certain populations can give important information about the kind of problem an individual suffers from. Oral microbiota alterations have been noted in dental caries, endodontic infections, gingivitis, and periodontitis [7] but also in more severe conditions. For instance, the prevalence of *Capnocytophaga gingivalis*, *Prevotella melaninogenica*, and *Streptococcus mitis* rises in patients with oral squamous cell carcinoma [8]. Surprisingly, tumors affecting distant organs may also induce a shift of the normal oral microbiota. In patients affected by pancreatic neoplasms, there is a lower presence of *S*. *mitis* and *Neisseria elongata* [9]. Diabetes was found to have correlations with oral microbiota alteration as well, even if the latter is a consequence of the pathology and not a contributing cause. Furthermore, the changes induced by the disease appear to be different depending on the diabetes type and original microbiota. However, the origin of the alterations may be the elevated glucose content of the oral environment and the compromise of the host immune response in diabetics [4,10]. Even cardiovascular diseases may be related to the oral microbiota, since bacteria typical of the oral environment were found in atherosclerotic plaques, their increased level being linked to higher cholesterol content in the blood of the patients [11]. Therefore, the oral bacterial microbiota can be useful to understanding the mechanisms triggering the onset of microbial diseases and facilitate the discovery of diagnostic markers [4].

Over recent years, some research has also been carried out in cattle, horses, and conventional pets, primarily to rule out differences between bacterial communities in healthy subjects and in animals suffering from oral diseases, such as periodontitis [12,13,14,15]. In healthy bovines, the most prevalent taxa are *Pseudomonas*, *Burkholderia*, and Actinobacteria [12]. In horses, *Gemella* and *Actinobacillus* seem to be the predominant genera when no oral disease is detected [13]. *Enhydrobacter, Moraxella*, and *Capnocytophaga* are the most abundant bacterial genera detected in healthy cats, and in dogs the most abundant taxa are an unclassified Pasteurellaceae sp., *Bergeyella*, *Conchiformibius*, and *Porphyromonas* [14,15].

In pig farming, a deeper look into the oral bacterial microbiota could lead to an improved management of economically detrimental diseases and be discussed for the gut bacterial microbiota [16]. According to Holman and colleagues, the gut microbiota consists mainly of the *Clostridium*, *Blautia*, *Lactobacillus*, *Prevotella*, and *Ruminococcus* genera. These bacterial genera may be benchmarks of a healthy gut microbiota in pigs [17]. Nevertheless, age is an important variable, since a significant shift in the gut bacterial microbiota was observed during the weaning period due to the transition to solid feed, which seems to produce a diversification in gut bacterial populations. Also, some anatomical sites are more subject to variations than others. In particular, the duodenum, jejunum, and ileum display more evident changes when compared to the cecum and colon [18]. Likewise, oral bacterial microbiota composition may be different depending on the anatomical site of sampling (tonsils, gingival surface, saliva, dental plaque) and on the age of the pigs under study. To date, few studies have investigated the oral bacterial community in pigs by means of 16S rRNA sequence analysis. These studies seem to confirm this theory, insomuch as the main components of the bacterial microbiota vary depending on the age and kind of sample. [5,19,20,21]. Furthermore, very little is known for some category of animals, such as sows; in fact, breeders offer a comprehensive picture of the bacteria that the pigs housed in the same herd could interact with. Therefore, the aim of the present study was to outline the bacterial microbiota of the oral cavity in sows.

## 2. Materials and Methods

### 2.1. Animals and Sampling Collection

Samples were collected from a herd of 70 Landrace × Large-White × Duroc hybrid sows and 3 Mora Romagnola boars located in Central Italy. Breeders and growers were vaccinated against Aujeszky’s disease (Aujeszky A-Suivax GI, FATRO S.P.A.), while sows were also vaccinated against *Escherichia coli* and *Clostridium* (*C*. *novyi* and *C*. *perfringens*) (SUISENG, Hipra). Piglets were vaccinated against porcine circovirus 2 (PCV2), *Mycoplasma hyopneumoniae*, and porcine reproductive and respiratory syndrome virus (PRRSV) at 3 weeks of age (3FLEX^®^, Boehringer Ingelheim) and weaned at 4 weeks. Weaned piglets were raised indoors until they reached a weight of 40–50 kg, when they were moved outdoors for the growing and finishing phase. Drinking water was taken from the potable public supply. Pigs were fed commercial grain-based rations composed of corn, barley, decorticated soybeans, wheat bran, wheat meal, soybean oil, calcium carbonate, dicalcium phosphate, and sodium chloride.

The most important infectious diseases in the studied herd are colibacillosis in sucklings, polyserositis in weaned piglets, and porcine dysentery in fattening pigs. Infection with *E*. *coli* was confirmed by bacteriological examination. The presence of *Brachyspira hyodysenteriae*, *Glaesserella parasuis*, *S*. *suis*, and *M*. *hyorhinis* was determined by molecular biological analysis. In particular, investigations were performed on tissue samples collected during post-mortem examination of naturally deceased pigs (data not published). Antibiotics such as gentamicin, amoxicillin, trimethoprim sulfamethazine, lincomycin, and tiamulin were used in the 2 years prior to the study to treat the above conditions. No antimicrobials were administered to sows during the sampling period, but probiotics and prebiotics (Very gut PRID547025, DSM nutritional products, Milan, Italy) were routinely added to the diet. The supplements contained mainly raw fiber, fatty acids, amino acids, and *Enterococcus faecium* NCIMB 10415 as the bacterial component.

Seven sampling sessions were conducted weekly from September to October 2020, and 22 saliva samples were collected in total from clinically healthy sows. There were 6 primiparous and 16 multiparous sows counted. Of the sows, 10 were gestating, 9 were lactating, and 3 were neither gestating nor lactating. A commercial kit containing a cotton rope, a sterile 50 mL tube, gloves, and a plastic bag was used to collect the samples according to the manufacturer’s instructions (PRRS Check by Unistrain, Hipra). Briefly, each rope was tied individually for 30 min to the cage of the sows, avoiding contact with the floor and the feeding fence. Some sows were excluded, because they showed no interest in the ropes. The ropes were picked up and squeezed into the plastic bag to extract saliva from the cotton threads. A tip of the bag was then cut off with clean scissors, and the contents of the bag were poured into the tube, which was labeled with the sow’s name and the day of sampling. Samples were all taken at approximately the same time of day, and animals were allowed to eat before, during, and after roping, as the herd was fed ad libitum. Samples were stored in an ice box for transport to the laboratory. There, they were aliquoted into 2 mL Eppendorf vials and stored at −80 °C until processing.

### 2.2. DNA Isolation and Sequencing

Library preparation and sequencing were performed at the DNA sequencing facility of the Life Sciences Department, Trieste University, Italy. Genomic DNA was extracted from 250 μL of saliva samples using the E.Z.N.A.^®^ Universal Pathogen Kit (Omega Bio-Tek, Norcross, GA, USA) according to the manufacturer’s instructions, eluted in 55 μL of elution buffer included in the kit, and stored at −20 °C until use. The quality and quantity of DNA was determined using a NanoDrop 2000 spectrophotometer (Thermo Fisher Scientific, Waltham, MA, USA). An extraction blank was performed as a control to monitor for contamination of environmental bacteria DNA. The extracted DNA was used as a template for amplification of the V4 hypervariable region of 16S rRNA using PCR primers 515F and a mixture of 802R and 806R [22,23]. The primers were tailed with two different GC-rich sequences enabling barcoding with a second amplification. Primary PCR amplification was performed in a 20 μL reaction volume containing 10 μL AccuStartII PCR ToughMix 2× (Quanta Bio, Beverly, MA, USA), 1 μL EvaGreen™ 20× (Biotium, Fremont, CA, USA), 0.8 μL 515 F (10 μM-5′ modified with Unitail 1 -CAGGACCAGGGTACGGTG-), 0.4 μL 802 R (10 μM-5′ modified with Unitail 2 -CGCAGAGAGGCTCCGTG-), 0.4 μL 806 R (10 μM-5′ modified with Unitail 2-CGCAGAGAGGCTCCGTG-), and up to 50 ng of DNA template. Amplification was performed in a CFX 96™ PCR system (Bio-Rad, Hercules, CA, USA) with a real-time limited number of cycles (94 °C for 20 s, 55 °C for 20 s, 72 °C for 60 s). A second PCR amplification (outer PCR) is required to uniquely label each sample and was performed with a forward primer composed of the ‘A’ adaptor, a sample-specific 10 bp barcode, and tail 1 of the primary PCR primers and a reverse primer composed of the P1 adaptor sequence and tail 2. Reactions were performed in a 25 μL volume containing 12.5 μL AccuStartII PCR ToughMix 2× (Quanta Bio, Beverly, MA, USA), 1.25 μL EvaGreen™ 20× (Biotium, Fremont, CA, USA), 1.5 μL barcoded primers (10 μM), 1 μL of primary PCR with the following conditions: 8 cycles of 94 °C for 10 s, 60 °C for 10 s, 65 °C for 30 s, and a final extension of 72 °C for 2 min. All amplicons were checked for quality and size by agarose gel electrophoresis, purified by Mag-Bind^®^TotalPure NGS (Omega Bio-Tek, Norcross, GA, USA), quantified by a Qubit Fluorometer (Thermo Fisher Scientific, Waltham, MA, USA), and pooled in equimolar amounts. The library was finally verified by agarose gel electrophoresis and quantified in a Qubit Fluorometer. For sequencing, the library was first subjected to emulsion PCR on the Ion OneTouch™ 2 system using the Ion PGM™ Template Hi-Q OT2 View (Life Technologies, Carlsbad, CA, USA) according to the manufacturer’s instructions. Ion sphere particles (ISP) were then enriched using the E/S module. The resulting live ISPs were loaded onto an Ion 316 chip in the Ion Torrent PGM system (Life Technologies, Carlsbad, CA, USA) and sequenced.

### 2.3. Data Analysis

The CLC Microbial Genomics Module, as part of the CLC Genomics Workbench 20.0 (QIAGEN Digital Insights, Aarhus, Denmark), was used to analyze alpha diversity and bacterial community composition. Raw sequencing reads were imported into the CLC environment, and we performed quality control, primers and adapters sequence removal, and a minimum size cut-off of 150 bp. OTUs were picked by mapping sequences against the SILVA 16S v13297% database [24] with the same identity percentage to observe OTUs at the species level. OTUs were then aligned by multiple sequence comparison using log-expectation and used to construct a “maximum likelihood phylogenetic tree”, followed by alpha diversity analysis. The graphical analysis was carried out in the R environment version 4.1.1 (R Core Team, version 4.1.1; R foundation for statistical computing: Vienna, Austria, 2021) [25,26,27,28].

### 2.4. Statistical Analysis

A permutational multivariate analysis of variance (PERMANOVA) test was applied to data to detect differences between gestating and lactating sows. Differences between genera and families representing ≥1% of the taxa were then investigated using Jamovi [29]. Data were compared using Student’s t-test or the Mann–Whitney U test, depending on the data distribution, which was previously checked using the Shapiro–Wilk test. The level of accepted statistical significance was *p* < 0.05.

## 3. Results

An average of 48,911–155,962 sequence reads of the V3–V4 region of the 16S rRNA gene were generated for each sample. In total, over 100 phylotypes were identified at the order level. The predominant orders belonged to Firmicutes (Lactobacillales, Clostridiales, Bacillales, and Erysipelotrichales), Actinobacteria (Corynebacteriales and Micrococcales) and Proteobacteria (Pseudomonadales) phyla. In particular, the Lactobacillales (30.2%), Clostridiales (13.5%), and Corynebacteriales (12.3%) accounted for more than 50% of the identified orders (Figure 1a).

In all of the samples studied, 241 families were identified, of which Lactobacillaceae (14.7%), Corynebacteriaceae (12.2), Moraxellaceae (10.0%), Aerococcaceae (6.9%), and Staphylococcaceae (6.2%) were the most represented. Together they accounted for approximately 50% of all families found in the samples examined (Figure 1b).

The most abundant genera were *Lactobacillus* (14.5%), *Corynebacterium* (10.3%), *Acinetobacter* (9.3%), *Staphylococcus* (4.4%), *Rothia* (4.2%), *Aerococcus* (4.0%), and *Clostridium* (3.9%), which accounted for more than 50% of the microbiota (Figure 1c).

Alpha diversity was measured using the Chao1, Shannon, and Simpson indices (Figure 2).

Principle component analysis (PCA) using the first two factors (PC1 and PC2) was performed using communities from each sample (Figure 3a,b). Each labeled point represents one sow community, while the numbers (1–58) represent the most represented genera (≥1% in at least one sample) (Appendix A).

As for the comparison between groups, statistically significant differences (*p* < 0.05) were found between pregnant and lactating sows. When further analyses were carried out on genera and families, statistically significant differences (*p* < 0.05) were found between pregnant and lactating sows. The first group had a higher number of Carnobacteriaceae and the second of Micrococcaceae. Even though not statistically significant (*p* > 0.05), other differences were particularly noted between gestating and lactating sows. Clostridiaceae, family XI -02, and Planococcaceae were more represented in pregnant sows, while Weeksellaceae, Staphylococcaceae, and Lactobacillaceae were higher in number in lactating animals (Figure 4a–c).

## 4. Discussion

Porcine bacterial microbiota is a research topic on the rise in veterinary medicine, although some anatomical areas have been more investigated than others. The reason may depend on the need to increase productivity and improve the health conditions of the herds. As for human medicine [7,8,9,10,11], the oral bacterial microbiota could be an instrument for the comprehension and prevention of current diseases, but it may also represent a research model for similar diseases in other monogastric animals, humans included. Available data on the oral bacterial microbiota of pigs are limited and differ from the results shown here. Regarding the orders identified as major components of oral communities in the present study, Lactobacillales, Clostridiales, and Corynebacteriales predominated. Most publications focus on comparisons between genera and do not describe in detail the major orders composing the oral bacterial microbiota of pigs. However, in the study by Lowe and colleagues, the order Pasteurellales alone was found to comprise 56% of the total bacterial microbiota of the tonsils of 12–16-week-old pigs [21]. In humans, the orders Lactobacillales, Clostridiales, Bacteroidales, and Enterobacteriales were described as the most prevalent [4], showing partial similarity with our results. Based on the obtained data, the most characteristic families in the samples studied were Lactobacillaceae, Corynebacteriaceae, Moraxellaceae, Aerococcaceae, and Staphylococcaceae. Studies of the tonsil microbiota identified the families Pasteurellaceae, Moraxellaceae, Fusobacteriaceae, Veillonellaceae, and Neisseriaceae in breeders [21] and Streptococcaceae, Staphylococcaceae, Micrococcaceae, Pasteurellaceae, and Moraxellaceae in 3-week-old piglets [20] as major components of the bacterial community. On the other hand, Streptococcaceae, Veillonellaceae, Prevotellaceae, and Enterobacteriaceae are widely distributed in the human oral bacterial microbiota [26]. In the present study, the most common genera were *Lactobacillus*, *Corynebacterium*, *Acinetobacter*, *Staphylococcus*, and *Rothia*. In their work, Murase and colleagues studied the salivary bacterial microbiota of piglets and discussed the genus composition of the oral cavity. The results of the study indicate that *Streptococcus*, *Moraxella*, *Actinobacillus*, and *Rothia* are the main genera composing the salivary bacterial microbiota of suckling piglets [5]. In humans, genera such as *Streptococcus*, *Neisseria*, *Gemella*, *Granulicatella*, and *Veillonella* have been identified as a possible core bacterial microbiota of the oral cavity, albeit with some variability [6,30,31,32,33]. With respect to the prevalence of some components over others in our results, *Lactobacillus* is not present in either the previously known oral core bacterial microbiota, or that of the tonsils [5,21]. The significant presence of the genus *Corynebacterium* has not been documented in the oral bacterial microbiota of pigs, and in human medicine its role has not yet been clarified, as it is an important component in dental biofilms but is more numerous in healthy individuals, thus playing a controversial role in oral wellbeing [34]. In general, the differences between the data from unrelated experiments are striking and could be due to factors such as the age and genetics of the pigs studied, the environmental conditions, the type of sample, the sampling method, the administration of pre- and probiotics, and the use of antimicrobial agents. In this case, probiotics probably did not directly impact on the oral bacterial microbiota composition, as *E*. *faecium* generally do not colonize the host when administered orally [35]. Nevertheless, *E*. *faecium* presence could have modified the bacterial arrangement within the oral cavity of sows as happens in the gut when such probiotics are given [36]. *E*. *faecium* properties as a probiotic in pigs were first investigated in 1984 and 1992, when it was administered to piglets in order to prevent or treat *E*. *coli* infection. Positive effects were demonstrated when *E*. *faecium* was used as preventive therapy, but it had no influence on the disease [37,38]. The route of action of *E*. *faecium* as a probiotic is still not clear; nonetheless, beneficial effects deriving from the interaction with the bacteria already present in the host may pay a significant role. When put together with *Lactobacillus* spp., *Salmonella enterica*, *E*. *coli*, and other strains or species of *Enterococcus*, *E*. *faecium* NCIMB 10415 achieves different results. Cultivation of *E*. *faecium* with some species of *Lactobacillus* enhances their growth, while it has no effect on others. *S*. *enterica* and *E*. *coli* can grow with *E*. *faecium* culture, but PCR exams revealed that the cell number of the pathogenic strains was lower than the control. Furthermore, it impairs the growth of some *Enterococcus* spp., depending on the production of bacteriocins by the co-cultured strain. In fact, in this case the probiotic strain’s growth is prevented. Therefore, it can be assumed that the action of *E*. *faecium* NCIMB 10415 on the microbiota is due both to the composition and the peculiar characteristics of bacteria living in the host [39]. In this particular case, the probiotic may have positively interacted with *Lactobacillus* species already settled in the sows, greatly enhancing their growth. Such a pattern would explain the differences between our results and the data reported in the aforementioned studies. Lactobacilli are widely recognized as positively correlated to a healthy microbiota, since they are capable of lowering the pH of the surrounding environment and hinder the growth of possibly pathogenic bacteria [40]. Moreover, through the same mechanism, lactobacilli may promote the diversity and the number of other indigenous lactobacilli, thus being considered good probiotics [41]. Even if still not proven, other mechanisms are thought to take part in interspecific stimulations, possibly due to growth promoting factors such as metabolites or bioactive substances [42].

Concerning the type of sample and collection method, cotton ropes were used in the present study to collect oral fluid. In pigs, oral fluids are usually collected either from one individual pig or a group of pigs by suspending a length of cotton rope in the pen then recovering the fluid by compressing the rope into a sterile container [43]. Oral sampling can be used to reveal the presence of antibodies [44] and for the detection of nucleic acids [45,46] or infectious viruses [47]. Pigs are curious animals, prone to exploration and oral manipulation of objects; thus, they are predisposed to bite and chew new elements introduced to their pen [48]. Oral fluid sampling using cotton ropes is voluntary and does not require animal restraint and is, therefore, considered welfare friendly [49]. Moreover, oral fluid sampling is easy to perform, requires minimal technical training, and is not time consuming [50,51]. Being suitable for nucleic acid detection, oral fluids can also be used more extensively to characterize the oral bacterial microbiota of individuals or groups of animals. As they are also suitable for the detection of nucleic acids, oral fluids have recently been shown to be useful to study the oral bacterial microbiota [52]. Indeed, Valeris-Chacin and co-authors used oral fluids to study the oral bacterial microbiota of pigs, relying on the quantification of *M*. *hyopneumoniae* and its correlation with community diversity and composition. However, the oral bacterial microbiota was not as well described as in the experiment by Murase and colleagues, whose work focused on quantifying *S*. *suis* rather than describing the composition of the bacterial communities [5]. To our knowledge, this is the first descriptive study of the oral bacterial microbiota of sows using cotton ropes for oral fluid collection. In addition, this study is based on a consistent number of samples from the herd (22 out of 70 sows). Nonetheless, it is compelling the part that the previous works played in defining the kind of sample and sampling sites housing most of the bacterial pathogens affecting the respiratory tract of pigs. Although the number of OTUs found in a sample is suggestive of the abundance of the genera, it does not give unambiguous identification of the species. Deeper bioinformatic or biomolecular investigations can provide such information. In any case, if a genus is highly represented in a sample, there is a strong possibility that among them there will also be pathogens belonging to that genus. Among the bacterial pathogens commonly involved in respiratory disease, the genera *Glaesserella*, *Pasteurella*, and *Mycoplasma* were found in minimal proportion and not in all samples. Conversely, *Streptococcus* represented 3.7% of the overall OTUs within the examined samples, but the species composing it were not investigated. To date, tonsils have been demonstrated to harbor potentially pathogenic bacteria, such as *G*. *parasuis*, *S*. *suis*, and *Pasteurella multocida* [21]. Conversely, *M*. *hyopneumoniae* has so far been encountered most frequently in tracheal fluid, while its concentration in oral fluid was generally undetectable. The reason for such a distribution could be ascribed to its tropism for tracheal cilia [52]. On the other hand, according to Murase and co-authors, saliva is the major habitat of *S*. *suis* [5]. It is worth pointing out that *S*. *suis* was endemic in the herd chosen for the present study, and it had been causing disease among the weaned pigs. Therefore, the presence of the pathogen in sows’ saliva may be consistent with the infection of the piglets, supporting the data set out by other authors.

Regarding the differences between pregnant and lactating sows, similar pre- and postpartum fluctuations have been observed in women [53], although the cause of these shifts is not yet clear. According to the most common management practice, sows adjust their diet after farrowing, while women generally do not drastically change their diet. Specifically, sows are fed a high-protein, high-fat diet around the time of farrowing and during lactation [54]. Restructuring of the gut bacterial microbiota after farrowing due to hormonal and nutritional adaptations has already been noted in pregnant and lactating sows [55]. In mice, pregnancy leads to an increase in genera such as *Clostridium*, *Akkermansia*, *Bacteroides*, and *Bifidobacterium* [56], and so a similar condition in sows could likely lead to changes as well. Remarkably, in the present case, such changes were not due to dietary variation, as the same diet was given to pregnant and lactating sows, confirming that other factors may contribute to bacterial microbiota alteration.

The technique utilized in the present study is accurate, but it has some limits. As shown in Figure 2 and Figure 4, mitochondria are included in the count of the most common families. In the present case, most of this genetic material was from *Solanum melongena* (eggplant) (Figure 3), *Avena sativa* (oat), and *Oryza sativa* subsp. *japonica* (Japanese rice). They could easily be excluded from the database, but it is interesting to note that feed remains should be considered when samples for oral bacterial microbiota analysis are taken from animals. In fact, in studies carried out on humans, people were asked not to eat for 1 h or wash their teeth even for 48 h before sampling [34]. Avoiding feed access for animals can be stressful or uneasy, especially when livestock is being examined. Moreover, the sampling method may have increased the quantity of feed fragments in the sample, ropes being likely to retain them while being chewed. Swabs and brushes allow a more accurate choice of the mouth area intended for the sample collection. On the other hand, similar techniques require animal restraint and possibly restlessness of the animals, thereby increasing the risk of contamination of the sample.

## 5. Conclusions

In the present study, the oral bacterial microbiota of sows was investigated. Briefly, Lactobacillales, Clostridiales, and Corynebacteriales were identified as the most represented orders, and Lactobacillaceae, Corynebacteriaceae, Moraxellaceae, Aerococcaceae, and Staphylococcaceae as the most numerous families. As regards the genus level, *Lactobacillus*, *Corynebacterium*, *Acinetobacter*, *Staphylococcus*, *Rothia*, *Aerococcus*, and *Clostridium* could be pointed out as the core bacterial microbiota. Sows were also divided into “gestating” and “lactating” groups, and mild differences were found between pregnant and lactating sows.

The data presented here aim to represent an important contribution to the knowledge of the porcine bacterial microbiota. Moreover, the choice of sows as experimental animals was strategic for identifying the adult microbial community. Indeed, sows represent the microbiological history of their farm and provide a comprehensive picture of the bacteria with which pigs belonging to the same herd might interact. These data provide a basis for further studies on the oral bacterial microbiota of pigs. More extensive studies, particularly based on larger sample size and closer control of diet and physiological conditions, will lead to a more thorough knowledge of oral communities and may have practical implications.

## Figures and Tables

**Figure 1 microorganisms-09-02314-f001:**
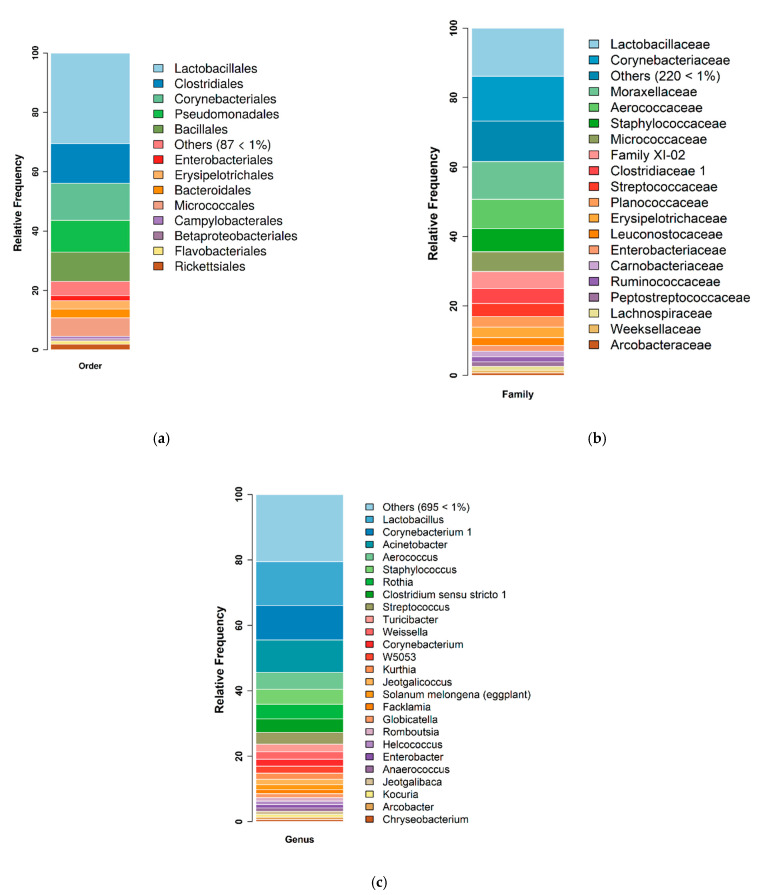
Bacterial microbiota structure of the oral fluid in sows (n=22). (**a**) Bacterial microbiota structure at the order level of the oral fluid in sows (*n* = 22). Only the bacterial orders that shared >1% abundance are indicated by different colors, while the others were grouped together. (**b**) Bacterial microbiota structure at the family level of the oral fluid in sows (*n* = 22). Only the bacterial families that shared >1% abundance are indicated by different colors, while the others were grouped together. (**c**). Bacterial microbiota structure at the genus level of oral fluid in sows (*n* = 22). Only the bacterial genera that shared >1% abundance are indicated by different colors, while the others were grouped together.

**Figure 2 microorganisms-09-02314-f002:**
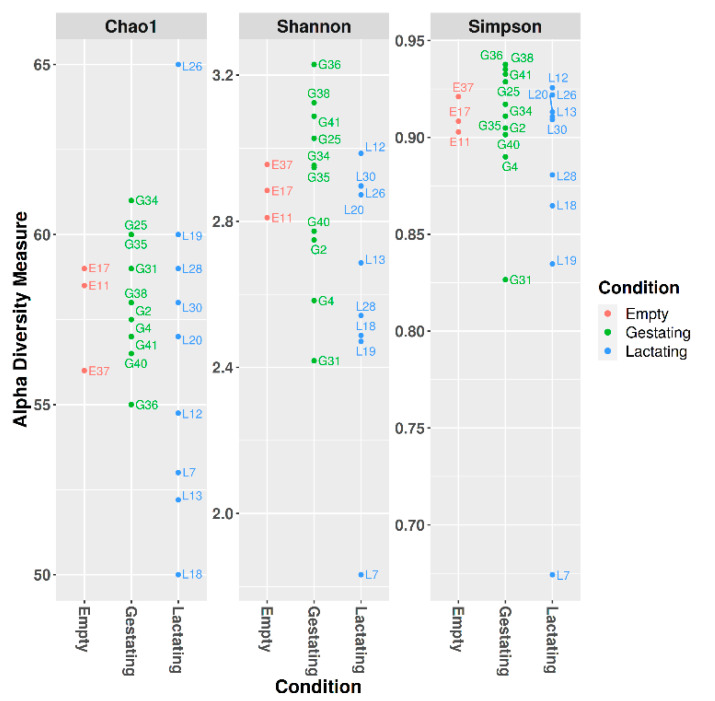
Alpha diversity based on the Chao1, Shannon, and Simpson indices.

**Figure 3 microorganisms-09-02314-f003:**
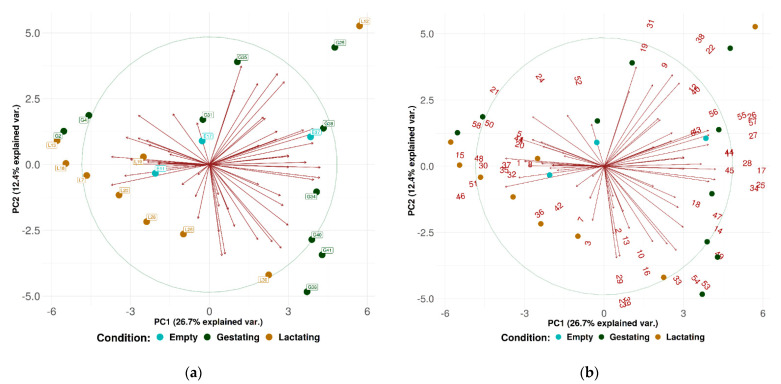
(**a**). Principal component analysis (PCA) characterizing the oral fluid bacterial microbiota of sows. Plot illustrating the distribution of the samples (*n* = 22) in the main two axes. (**b**) Principal component analysis (PCA) characterizing the oral fluid bacterial microbiota of sows. Plot illustrating the distribution of the main genera (*n* = 58) in the main two axes.

**Figure 4 microorganisms-09-02314-f004:**
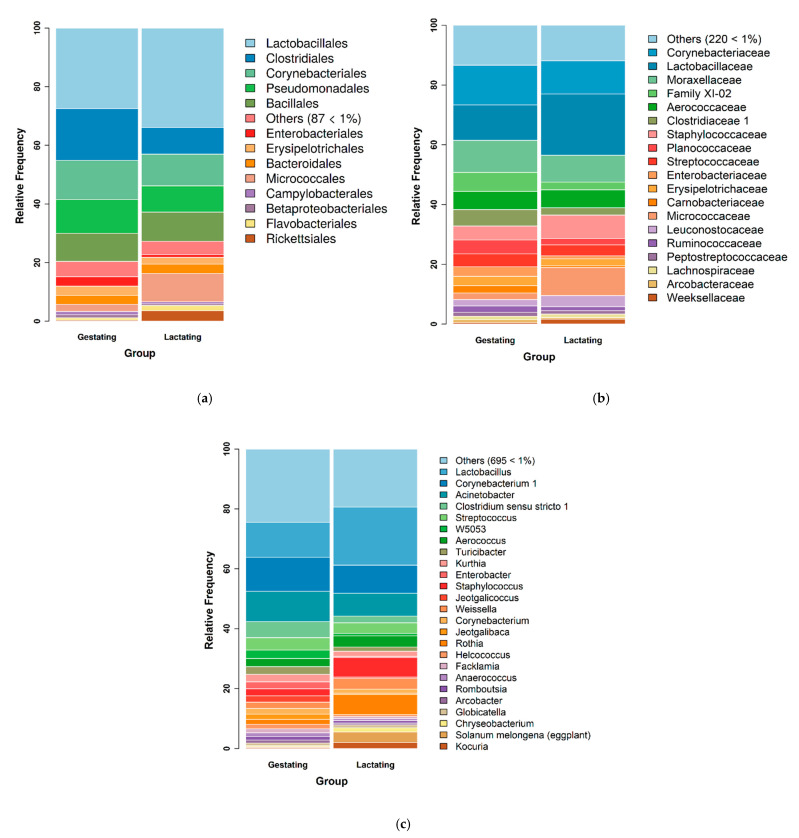
Bacterial microbiota structure of the oral fluid in gestating (*n* = 10) and lactating (*n* = 9) sows. (**a**). Bacterial microbiota structure at the order level of the oral fluid in gestating (*n* = 10) and lactating (*n* = 9) sows. Only the bacterial orders that shared >1% abundance are indicated by different colors, while the others were grouped together. (**b**) Bacterial microbiota structure at the family level of the oral fluid in gestating (*n* = 10) and lactating (*n* = 9) sows. Only the bacterial families that shared >1% abundance are indicated by different colors, while the others were grouped together. (**c**) Bacterial microbiota structure at the genus level of the oral fluid in gestating (*n* = 10) and lactating (*n* = 9) sows. Only the bacterial genera that shared >1% abundance are indicated by different colors, while the others were grouped together.

## Data Availability

All data are contained within the manuscript.

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
