# Peer review of "Insights into the Oral Bacterial Microbiota of Sows"

_microorganisms, 2021, doi:10.3390/microorganisms9112314_

Round 1

Reviewer 1 Report

Accept in present form.

Author Response

The authors thank the Reviewer for his/her kind report.

Reviewer 2 Report

Since there are not much work address the saliva microbes in pigs, the scientific soundness of manuscript is critical. It is great to see author’s effort to improve data presentation. Please clarify the samples collection. As my understanding, there were a total of 22 samples collected at each time point and each sow was collected seven times (weeks) repeatedly, right? In this case there were 154 samples subject to 16S RNA sequencing, correct? I can relate to gestation sampling but am confuse on lactation sampling since lactation period typically last for 4 weeks? Please define the 3 empty sows. What is parity of three empty sows and why are they empty? Also of those 22 samples how many of those are from primiparous and how many are from multiparous? What is the housing system for sows?

After these questions are answered, I would be happy to review the manuscript.

I do notice that offspring were kept through market. Is there offspring data available?

Author Response

Please, find the file herein attached.

Round 2

Reviewer 2 Report

After further review, the materials content in this work is okay to publish as research note but not sufficient to publish as a full publication. 

Author Response

Dear Reviewer,

We accept to change the format as "research note" (Communication, as indicated by the Academic Editor).

Regards,

Pietro Giorgio Tiscar & co-Authors

Round 3

Reviewer 2 Report

Please combine figure 1a, 1b and 1 c like you do for figure 2 as well as figure 4a, 4b and 4c